# Justice at Risk? The Influence of Recidivism Risk Information on Evaluation of Evidence and Determination of Guilt

**DOI:** 10.3390/bs15091277

**Published:** 2025-09-18

**Authors:** Tamara L. F. De Beuf, Roosmarijn M. S. van Es, Jan W. de Keijser, Henry Otgaar

**Affiliations:** 1Netherlands Institute for the Study of Crime and Law Enforcement, 1008 BH Amsterdam, The Netherlands; 2Faculty of Law and Criminology, KU Leuven, 3000 Leuven, Belgium; 3Institute of Criminal Law and Criminology, Leiden University, 2311 ES Leiden, The Netherlands; 4Faculty of Psychology and Neuroscience, Maastricht University, 6211 LH Maastricht, The Netherlands

**Keywords:** context effect, recidivism risk, legal decision-making, evidence evaluation

## Abstract

In contrast to jurisdictions with bifurcated criminal justice proceedings, in Belgium and the Netherlands a defendant’s assessed risk of recidivism is known to triers of fact prior to making decisions about guilt. In three experiments conducted in those two countries, we investigated whether information about recidivism risk would bias the fact finders’ evaluations of evidence and the defendant’s credibility, and their final decision on guilt. Specifically, student participants (Belgian sample: *N* = 368; Dutch sample: *N* = 236) and jury-eligible Belgian participants (*N* = 75) read a vignette about an aggravated assault with circumstantial evidence and a defendant who denied committing the alleged offense. Participants were randomly assigned to one of three vignettes: one without risk information, one describing a low risk of recidivism, and one describing a high risk of recidivism. We found no direct or indirect effect of risk on the proportion of guilty verdicts or on the evaluation of the evidence. We did find that participants who read that the defendant was low risk evaluated the innocence claim as being more credible, compared to those who were given high-risk information or no risk information. Moreover, higher credibility ratings were associated with a higher likelihood of a not-guilty verdict. While preliminary, these findings suggest recidivism risk information may influence fact finding, and merit replication, especially with judges.

## 1. Introduction

Risk assessment has become an essential part of criminal justice decision-making ([54]). Judges use risk assessment information when making decisions, both before and after the conviction of a defendant. In pre-trial settings, risk assessment can help in estimating the likelihood that a defendant will fail to appear at a scheduled court hearing or will commit new offenses while awaiting trial. It can, therefore, influence the decision on pretrial detention, in addition to the other pieces of information that judges consider. Additionally, judges consider an individual’s risk of recidivism when determining the severity and length of a sentence, as well as whether continued incarceration is necessary to protect society ([28]; [29]; [53]). Overall, the legal decision-maker must balance public safety with the person’s legal and human rights ([8]). These evaluations are guided by the outcomes of (structured) risk assessment instruments applied by trained professionals. These professionals can be probation, parole, and law enforcement officers, or forensic mental health professionals such as forensic psychologists, forensic psychiatrists, nurses, and social workers ([43]).

Violence risk assessment involves estimating the likelihood that an individual will engage in future harmful, violent behavior ([15]). There are various approaches to conducting these assessments, ranging from unstructured clinical judgment—where assessors rely solely on their experience and intuition—to structured methods that incorporate evidence-based risk factors ([15]; [52]). Regardless of the approach, risk assessment has become a critical tool in the criminal justice system, shaping decisions on pre-trial detention, sentencing, release, civil commitment, and even death penalty cases ([13]). The present study examined whether risk assessment information may bias other legal decisions in criminal courts, in particular, those decisions for which risk is not a relevant factor. Specifically, we examined whether receiving information about recidivism risk would affect the evaluation of evidence, the defendant’s credibility, and the final decision on guilt. While some scholars argue that risk information is also irrelevant to sentencing—contending that punishment should focus on retribution based on past behavior—we acknowledge the widespread practice in Western jurisdictions of incorporating risk assessments into sentencing decisions (for a discussion, see [44]).

### 1.1. Contextual Bias

Bias can happen when information has an unjustified impact on a decision-making process, potentially resulting in incorrect decisions ([23]). One common type of bias is confirmation bias, which happens when pre-existing opinions, beliefs, and expectations influence how someone seeks, selects, and interprets information ([31]). For example, once the police have a suspect for a particular crime, they tend to exclusively search for more incriminating evidence relating to that specific suspect, while ignoring exculpatory evidence ([37]). This is also referred to as belief-consistent information processing, and it is argued that it is fundamental to a wide variety of cognitive biases (see [33]). One specific type of confirmation bias is contextual bias, in which context-based information (e.g., prior information) influences information processing and decision-making. For example, crime scene investigators have been shown to interpret a crime scene differently with respect to securing trace evidence, depending on whether someone told them prior to their assessment of the crime scene that it was either a suicide or a violent death (i.e., contextual information; [49]). This type of (forensic) contextual confirmation bias has been demonstrated across various forensic disciplines (see [24]). In the current study, we evaluated whether risk information, as a specific type of contextual information, can bias legal decision-makers when they make judgments for which this information is irrelevant.

### 1.2. Risk Information and Legal Decisions

While recidivism risk is relevant to decisions on sentencing and pre-trial detention, it should be irrelevant in the fact-finding phase of a trial, in which the evidence is considered and a defendant’s guilt is determined. Decisions made in this trial phase concern past actions, whereas recidivism risk is about assessing the risk of future criminal behavior, which is supposed to be especially relevant for decisions about treatment and sanctions. These are fundamentally different inquiries, and hence, risk assessment reports should not be used during the fact-finding phase, as they could potentially influence verdicts.

Some countries have implemented procedural rules to ensure that triers of fact do not have risk information prior to the sentencing phase, to prevent this information from affecting the verdict. In the US, for instance, risk information may only be used after a guilty verdict, at the sentencing stage, to inform decisions about appropriate sanctions, since risk information is used to predict future behavior and not as evidence of past behavior. Risk information is therefore typically not presented at the guilt stage. Such a bifurcated or two-stage trial does not exist in countries such as Belgium and the Netherlands (e.g., [17]). In these countries, the information about future criminal risk is admissible and available to triers of fact from the start of the trial, because it is included in the court file, often as part of a forensic mental health report or a presentence report. Moreover, in Belgian cases, in the court of assizes, in which the most serious offences are tried by a jury, forensic mental health experts are called to testify about their reports before the jury renders a verdict ([48]). Their expert-witness testimony also involves a risk assessment which is later used by the jury and the three presiding judges to inform their decision-making during the sentencing phase of the trial. Similar procedures are in place in the Netherlands, although Dutch courts exclusively consist of proceedings before professional judges, and it is not mandatory or common practice to have experts testify in court. The expert reports are included in the case file that is available before the start of the trial.

There is preliminary evidence that recidivism risk information presented in an expert report may (negatively) affect legal decisions prior to the sentencing phase. One experimental study found that risk information influences the effects that Dutch forensic psychological reports have on verdicts ([50]). Specifically, the presence of a forensic psychological report increased the proportion of guilty verdict by almost 20%, irrespective of the reported psychological disorder. Interestingly, when recidivism risk information was included in the report, this biasing effect was diminished. When risk information was present, the proportion of guilty verdicts (80%) was significantly *lower* than when there was no information about risk in the report (94%). There were no statistically significant differences between low and high risk. The researchers argued that the risk information attenuated the biasing effect of the mental health report. Regardless of potential explanations, the presence of risk information seemed to have influenced the decision-maker, even though this information should be irrelevant at this stage of the court process. Prior research has shown that judges and jurors can struggle to disregard irrelevant information, even when explicitly instructed that such information was inadmissible ([46]; [57]). However, to our knowledge, no studies have specifically examined whether this inability to disregard irrelevant information extends to risk assessment information.

This finding raises questions about the potential direct effect of risk information on verdicts, that is, when it is not part of a broader forensic psychological report including information about mental disorders. Through the mechanism of contextual bias, information about a defendant’s high risk of committing future offenses may increase the likelihood that legal decision-makers will render a guilty verdict ([12]; [51]). Information about a high risk of recidivism may trigger a ‘guilt schema’ in the decision-maker’s mind; a mental representation of a guilty person is activated ([7]; [53]). Subsequent processes, such as interpreting the evidence and rendering a verdict, may become biased because of this schema. Especially in more ambiguous cases, with a defendant who denies committing the alleged offense, and relatively weak evidence, triers of fact may be looking for additional information for their decision-making and resort to information that is in fact irrelevant in this trial phase, such as information about recidivism risk ([21]). The influence of risk information may alternatively occur via the horn effect, also known as the reverse halo effect, a cognitive bias that results in an overall negative evaluation of a person based on one specific negative characteristic ([32]). Information about the person’s high risk may transpose to an overall negative judgement of their character, resulting in the triers of fact considering this person to be someone who commits crimes.

### 1.3. Risk Information and Contamination of Evidence

Irrelevant contextual information may also indirectly impact the determination of guilt. It may affect the evaluation of the evidence in a case. In a series of experiments with students and police officers, [4] ([4], [3], [5]) demonstrated that an initial belief of guilt influenced the evaluation of a piece of evidence, for example, an alibi. This biased evaluation then significantly influenced the evaluation of another piece of evidence. This mechanism was also shown in experiments by [36] ([36]). Here, knowledge of incriminating DNA evidence inflated the estimated strength of subsequent eyewitness-identification evidence, as well as the overall conviction of guilt and increased the likelihood of a guilty verdict. This mechanism is known as the ‘bias snowball effect’, or ‘assimilation of evidence’ [16]; [36]). A similar effect was also found for information that should not be relevant to the conviction, such as information about the suspect’s psychopathology. Knowledge of this information inflated the estimated strength of subsequent fingerprint evidence and—as a result—the likelihood of a guilty verdict ([36]). In sum, the initial belief of guilt, combined with biased interpretation of evidence, can strengthen the belief that the suspect is guilty. This reflects a form of confirmation bias driven by people’s tendency to seek consistency and coherence in their reasoning processes—also referred to as coherence-based reasoning ([42]). To date, it has not yet been empirically examined whether the availability of recidivism risk information has a similar snowballing- or assimilation-effect. In other words, it is not known whether knowledge about a defendant’s risk of future violent behavior influences the evaluation of evidence, which may in turn influence the judgment of guilt.

### 1.4. Relevance in Plea Negotiations

We were interested in the potential context effects of risk information on legal decision-making in the absence of relevant procedural rules, as is the case in Belgium and the Netherlands. While procedural rules in jurisdictions like the United States may actively prevent the unwarranted effects of risk information on verdicts, this research question may still be relevant in the context of American plea bargaining, in which prosecutors and defendants negotiate an agreement. During plea negotiations, risk information is sometimes available and used by prosecutors and defense attorneys ([27]). Thus, the availability of risk information could potentially affect plea deals, for example, if a high risk of recidivism is considered to be an indicator of guilt, and subsequently the defendant is urged to plead guilty. Studying this issue is particularly relevant, because around 90% of federal convictions in the U.S. result from guilty pleas, with the majority—about 75%—emerging from plea negotiations ([22]; [19]). This underscores the need to understand how factors like recidivism risk might influence judgments, even outside the trial setting. Although the penalties following guilty pleas are usually less severe than after a trial, the consequences can still be profound (e.g., criminal record, prison sentence, loss of job, and deportation; [59]), especially when an innocent person has been urged to plead guilty.

### 1.5. Risk Information and Suspect Statement Credibility

A related question is whether recidivism risk information affects the perceived credibility of a defendant’s claim of innocence, and if so, in which direction. Research has shown that the credibility of a defendant and/or of their statements can be influenced by irrelevant information such as the defendant’s emotional presentation, appearance, or nonverbal behavior. For example, [21] ([21]) demonstrated that defendants who showed little visual or vocal emotion were perceived as being less credible than highly emotional defendants, but only when the evidence was weak. This finding was replicated by [55] ([55]), who further found that the valence of the emotion mattered. Specifically, defendants who expressed sadness and despair were perceived as being more credible than those who were more lighthearted. Other irrelevant contextual information has also been demonstrated to affect suspect statement credibility. A recent study examined whether the direction of the evidence, incriminating versus exonerating, affected alibi credibility ([39]). Participants who received incriminating evidence judged the suspect’s alibi statement to be less credible, compared to those who received exonerating evidence. Also, nonverbal behaviors that are typically but unduly linked to deceit, such as gaze aversion or fidgeting, can affect credibility assessment. [2] ([2]) found that a defendant’s statement was considered less credible when more of such stereotypical nonverbal behaviors were observed in the fictive defendant.

In sum, there is preliminary evidence that irrelevant contextual information affects the perceived credibility of a defendant and of their statements. Could information about a defendant’s level of recidivism risk have a similar biasing effect on the assessment of the credibility of a defendant’s statements and, in particular, their claim of innocence? Again, the assessment of high risk may invoke stereotypical thinking about the defendant as a dangerous perpetrator and subsequently reduce the credibility of their claim of innocence. To our knowledge, there have been no studies to date that have evaluated the impact of risk assessment information on the perceived credibility of a defendant’s claim of innocence.

### 1.6. The Present Experiments

The present experiments examined whether the availability of risk information would affect decision-making, and whether this would depend on the level of risk (low or high) that was assigned to the defendant. More specifically, we hypothesized that information about recidivism risk would lead to more guilty verdicts, compared to the absence of risk information, but only among participants who read that the defendant was at high risk of recidivism. We did not expect any effect of risk information on the verdicts among participants who read a vignette of a low-risk defendant. We, therefore, expected no differences in terms of the proportion of guilty verdicts in the low-risk condition versus the control condition. Our hypotheses were inspired by the general finding of [50] ([50]) that the mere presence of risk information in a forensic mental health report affected the conviction rate. Additionally, the present experiments tested whether the reported level of recidivism risk mediated the effect on judgments of guilt via the evaluation of evidence. We hypothesized that recidivism risk information would influence the evaluation of the presented evidence, which would in turn affect how convinced one would be of the defendant’s guilt, as well as whether one would render a guilty verdict. Lastly, we tested whether the reported level of risk influenced the perceived credibility of a defendant’s claim of innocence. We expected that participants would perceive the defendant’s innocence claim as less credible when they read that the defendant was high risk compared to low risk. This hypothesis was based on studies demonstrating the influence of irrelevant contextual information on credibility (e.g., [2]; [39]; [55]).

## 2. Method

Our first experiment included students at the Faculty of Law and Criminology at KU Leuven, Belgium (Experiment 1). Subsequent to this experiment, two follow-up experiments were conducted: one with students at the Leiden Law School at Leiden University in the Netherlands (Experiment 2) and one with jury-eligible Belgians (Experiment 3). The approaches of all three experiments were similar, using the same case vignette, design, and procedure. There are, however, some key differences in terms of the dependent and independent variables, which will be discussed below.

### 2.1. Case Vignette

After giving informed consent, participants read a case vignette about an aggravated assault in which the defendant maintains his innocence. The case vignettes were adapted from the study by [50] ([50]), which in turn were based on a fictitious but realistic case file designed by [9] ([9]). The vignettes describe a defendant who goes out and has several beers with two friends. After leaving the bar, they run into the victim and his girlfriend, who are passing by. A remark from the defendant to the girlfriend results in an argument between the defendant and the victim. Thedefendant allegedly followed the victim and his girlfriend and attacked the victim. The victim was kicked in the chest and beaten multiple times, until he did not move anymore. The attack resulted in severe consequences for the victim: loss of speech, permanent paralysis, and memory loss. Other than the girlfriend, no one witnessed the assault. The defendant’s two friends went home right after the altercation outside the bar. When interviewed by the police, the defendant maintains his innocence and states that he is not the person who assaulted the victim. He claims that he went straight home after the altercation. The case vignette included the summary of two suspect interrogations, witness testimonies of the defendant’s two friends as well as the victim’s girlfriend, a hesitant identification by the girlfriend of the defendant in a photo line-up, and an expert report on the victim’s injuries. The evidence against the defendant consisted therefore of the two pieces of testimony by witnesses (one from the defendant’s friends, one from the victim’s girlfriend) and the girlfriend’s line-up identification. The evidence in the vignette was purposefully circumstantial and inconclusive across all experimental conditions.

In the original vignette, [9] ([9]) created ambiguity about the defendant’s guilt to allow for their manipulation to have an effect.[note 1] They argued that evidence that was too weak or too strong would result in the overwhelming majority of judges, if not all, rendering the same verdict, not guilty or guilty, respectively, which would make it impossible to test the hypothesis. Although our hypotheses are different from those of de Keijser and van Koppen, this approach applies to our study, in that doubt is required in order to be able to measure a potential context effect. That is, research has shown that a context effect is most likely to appear in cases with ambiguous evidence ([11]; [21]).

The original Dutch vignette was adjusted for the Belgian context, using Flemish terminology (e.g., the job titles of criminal-justice actors), names, and locations. Although this implies that the vignettes of Experiments 1 and 3 used slightly different terminology compared to the vignette in Experiment 2, it is unlikely that this affected the narrative. Moreover, the experimental manipulation was identical for both contexts, except that in Belgium, an expert witness completed the risk assessment, and in the Netherlands, it was conducted by probation services. After reading the vignette, participants were asked to render a verdict.

### 2.2. Recidivism Risk Information

In the control condition, no recidivism risk information was provided. Recidivism risk information was included in the two experimental conditions. In one condition, the defendant was assessed as having a high recidivism risk, and in the other condition, the defendant was described as having low risk of recidivism. In Experiment 1, the recidivism risk information consisted of simplified summary of the final risk judgment based on the assessment of the Historical Clinical Risk management–20 Version 3 (HCR-20^V3^; [14]). The paragraph described in general terms some of the risk factors that are included in the instrument and was identical for both experimental conditions, except for the concluding risk judgment: “*Based on the 20 items of the HCR-20 V3*, *the level of risk for future recidivism of the defendant is estimated to be [low/high]*.” The case vignettes with risk information consisted of 1262 words, and the (control) vignette without this information (i.e., control condition) was 1126 words.

To improve the ecological validity of the manipulation, the reported risk information was personalized to the defendant in Experiments 2 and 3. That is, the risk paragraph described which specific risk factors contributed to the final risk judgment of a low, versus high, risk:

*The risk assessor*[note 2] *made a risk assessment for future violent behavior using the structured clinical instrument called Historical Clinical and Risk Management-20 Version 3 (HCR-20 V3). This instrument consists of 20 factors that describe various characteristics of the assessed person through an evaluation of one’s background and living situation. The evaluation revealed that Martens dropped out of vocational training [Low risk: finished high school] and is unemployed [Low risk: was first unemployed for a while*, *but for the past year he has had a permanent job in construction]. At the age of 17*, *he ran away from his parents and has since been couch surfing at various friends’ places [Low risk: Martens still lives with his parents]. At the time of the risk assessment*, *Martens was not in a relationship; his dating life mainly consists of short*, *fleeting relationships [Low risk: he was no longer in a relationship but still had good and supportive contact with his ex-girlfriend]. Additionally*, *he frequently smokes marijuana and out of boredom he often hangs out on the streets with friends who cause trouble in the neighborhood. Martens is hot-tempered but has not previously come into contact with the criminal justice system [Low risk: Martens has no history of drug use and has not previously come into contact with the justice system. Martens comes across as a calm young man*, *which makes it difficult for the risk assessor to understand why he went mad like that]. The risk assessor concludes that the risk of Martens committing a violent crime in the future is high [low].*

The vignettes with risk information for Experiments 2 and 3 consisted of approximately 1270 words, and the vignette without risk information (control) included approximately 1100 words. There were minimal differences in total word count between the Dutch and Belgian version. All vignettes, in addition to their English translations, are accessible via OSF (https://osf.io/avz4h/).

### 2.3. Design and Procedure

All experiments followed a between-subjects design with level of risk information as the independent variable (3: no risk, low risk, and high risk). Participants were randomly assigned to the three conditions, using the randomizer built into survey tool Qualtrics.

All experiments included attention checks. These attention-related questions were meant to check participants’ engagement with the material and whether they had read the information with sufficient care. Participants had to answer two questions about the witness’ testimonies, two questions about the suspect’s interrogations, and one question about the expert report. Together with the last question, participants were asked about the defendant’s risk of recidivism: “*What was Mr. Martens risk of future violence as assessed by the HCR-20 V3?*”. The options for the answer were “not provided”, “low risk”, and “high risk”. (This was a manipulation check to check whether participants had read and understood the parts central to the manipulation.) The attention and manipulation checks were asked on separate pages of the online survey, without the option to return to the case information. As defined in the preregistration, participants who failed multiple attention checks (Experiment 1: two out of five, Experiments 2 and 3: three out of five) and/or the manipulation check were excluded.

Across all experiments, participants were asked to decide, based on the evidence and their level of conviction, whether the defendant was guilty of the assault (Experiment 1; dichotomous guilty/not guilty) or should be convicted (Experiments 2 and 3; dichotomous yes/no). In addition to this central question, participants in Experiment 1 were asked how credible they found the defendant’s claim of innocence, as rated on a scale of 1 (*not credible at all*) to 10 (*very credible*). Participants in Experiments 2 and 3 received other additional questions. That is, they were asked to indicate how incriminating the witness statements in the vignette were for the defendant on a scale of 0 (*not incriminating at all*) to 10 (*very incriminating*), and how incriminating the identification from the line-up was on a scale of 0 (*not incriminating at all*) to 10 (*very incriminating*). Additionally, in Experiments 2 and 3, participants indicated how convinced they were of the defendant’s guilt on a scale of 0% (*not at all convinced*) to 100% (*totally convinced*).

The final questions of the survey related to demographics: gender, age, level of education (Belgium), whether participants professionally came into contact with the criminal justice system (Belgium), year of study (Netherlands), field of study (Netherlands), and university (Netherlands). Level of education (Belgium), experience with the criminal justice system (Belgium), and year of study (the Netherlands) were assessed to be used in the analyses as covariates. We expected that these variables would affect the dependent variables, because they can be associated with knowledge about and affinity with the criminal justice system. The last phase of the survey was a debriefing.

### 2.4. Data Analysis

First, to double-check randomization, we assessed, as preregistered, whether there were statistically significant differences between the conditions (low risk, high risk, and no risk). Using a chi-square test, differences were examined for the field and year of study, level of education, experience with the criminal justice system, and job situation, as well as gender. Age differences were tested using independent t-test, or the Mann–Whitney test if the assumption of normality was violated.

To assess the relationship between the risk groups and the proportion of guilty verdicts, a chi-square test was conducted in all experiments. To examine whether risk assessment information resulted in different credibility ratings for the defendant’s claim of innocence (Experiment 1), a one-way ANOVA was conducted. A post hoc Games–Howell *t*-test compared the three conditions in order to assess which ones differed significantly. The correlation between the dependent variables of verdict and credibility was analyzed using a point biserial correlation. In addition to these preregistered analyses, we conducted an exploratory *t*-test to compare the credibility ratings of participants who rated the defendant as guilty versus those who rated him as not guilty.

To test the effect of recidivism risk on the evaluation of the two pieces of evidence, one-way ANOVAs (Experiment 2) and a MANOVA (Experiment 3) were conducted, whereas one-way ANOVA was conducted to assess the effect of recidivism risk on the participants’ conviction of guilt (Experiments 2 and 3). In Experiments 2 and 3, a (logistic) regression analysis was conducted to explore the impact of the level of recidivism risk and evaluation of evidence on the defendant’s conviction of guilt and likelihood of the defendant being convicted. The analyses were conducted using JASP 0.17.3.0 and SPSS 29.

## 3. Experiment 1

Ethical approval for Experiment 1 was granted by the Social and Societal Ethics Committee of KU Leuven (reference: G-2022-5226). The experiment was preregistered[note 3] on the Open Science Framework (OSF; https://osf.io/ce4u7) and all materials and data are available at OSF (in Dutch; https://osf.io/avz4h/).

### 3.1. Participants

An a priori power analysis was conducted to calculate the required sample size for a chi-squared (goodness-of-fit) test. Using G*Power version 3.1.9.7 (see OSF; [18]), with a power of 0.80, α = 0.05, *w* = 0.18, and two degrees of freedom, we calculated that a sample of 298 participants was needed. We based the Cohen’s *w* effect size on the effect that was found by van Es and colleagues (2022) when they examined the impact of recidivism risk information in pre-sentence reports on verdicts. They found φ = 0.18, and for 2 × 2 contingency tables this can be translated into *w* = 0.18. Although not all the analyses in this study were 2 × 2, this effect size was used as an estimate. To account for attrition, we aimed to collect data from 400 participants.

Participation was voluntary. Participants were recruited in class during the courses Criminological Psychology and Legal Psychology at KU Leuven’s Faculty of Law and Criminology. During lectures and via announcements on the university’s online learning environment, students were invited to participate in the experiment. The study fully took place online using Qualtrics, and the link was shared during the in-class and online recruitment. At the beginning of the lecture, 15 to 20 min were reserved to complete the survey, and the link remained available for several weeks for students to participate at a later time. Students were not reimbursed for their participation.

In total, 574 participants started the Qualtrics survey and 520 participants completed the survey. Of those, almost one-third (*n* = 149, 28.7%) failed the manipulation check and/or two or more attention checks. In addition, three students did not study criminology or law, and were excluded. The final sample consisted of 368 participants, mainly female students (*n* = 279, 75.8%; male: *n* = 85, 23.1%; other: *n* = 4, 1.1%) who studied law (*n* = 249, 67.7%; criminology: *n* = 119, 32.3%). About half of the sample were graduate students (i.e., master’s degree students; *n* = 200, 54.3%) and the other half were undergraduates (*n* = 168, 45.7%). Due to a mistake relating to the software, age information was not gathered. The group that did not receive risk information consisted of 123 participants, the group with information about a low recidivism risk of 124 participants, and the group that received information about a high risk of recidivism consisted of 121 participants.

### 3.2. Results

There were no statistically significant differences between the experimental conditions in terms of gender, field of study, and year of study. Across all conditions, 74.2% found the defendant guilty of the aggravated assault (*n* = 273). In line with the experiments of [9] ([9]) and of [50] ([50]), this conviction rate indicates that the evidence in the case was ambiguous enough to cast some doubt on the defendant’s guilt.

We compared the decisions about guilt among the three groups that received different risk information. In the group that received no risk information (*n* = 123), 73.2% of the participants (*n* = 90) judged the defendant to be guilty. In the group that read that the defendant had received an assessment of a low risk of future violence (*n* = 124), 70.2% judged that he was guilty (*n* = 87), and among those who read he was high risk (*n* = 121), 79.3% of the participants rendered a guilty verdict (*n* = 96). A chi-square test demonstrated that these differences in the proportion of guilty verdicts were not statistically significant (*χ*^2^ (2) = 2.793, *p* = 0.248, Cramer’s *V* = 0.09). As an exploratory (not preregistered) analysis, odds ratios were calculated in order to provide insight into the strength of the effects, even though they are not significant. Participants in the high-risk group were slightly more likely to receive a guilty verdict than those in the group with no information about risk (OR = 1.41, 95% CI [0.78, 2.55]) and those in the low-risk group (OR = 1.63, 95% CI [0.91, 2.93]). Participants in the low-risk group were slightly less likely to award a guilty verdict, compared to those in the group with no information about risk (OR = 0.86, 95% CI [0.50, 1.50]).

Using one-way ANOVA, we examined whether the groups statistically differed as to how credible they found the defendant’s claim of innocence. Since the variances were not equal, the Welch homogeneity correction was applied. We found a statistically significant difference in credibility between at least two groups (Welch’s *F*(2, 239.26) = 10.39, *p* < 0.001, *η*^2^ = 0.064). A Games–Howell test for multiple comparisons of means with unequal variances found that the mean credibility rating was statistically significantly different between the low-risk group (*M* = 4.22, *SD* = 1.89) and the high-risk group (*M* = 3.28, *SD* = 1.31; *p* < 0.001, *d* = 0.58). In absolute terms, this means that the low-risk group rated the credibility of the innocence claim on average almost one point higher on the Likert scale than the high-risk group did. In relative terms, 71.9% of the low-risk group rated credibility higher than the average credibility rating of the high-risk group, and there is a 65.9% chance that a person picked at random from the low-risk group would be associated with a higher credibility rating than a person picked at random from the high-risk group ([26]).

There was also a statistically significant difference between the low-risk group and the control group (*M* = 3.50, *SD* = 1.36; *p* = 0.002, *d* = 0.44). The absolute mean difference is somewhat smaller than in the previous comparison: the low-risk group rated the innocence claim, on average, 0.72 points higher than the control group. Moreover, 67.0% of the low-risk group rated credibility higher than the average credibility rating of the control group, and there is a 62.2% chance that a person picked at random from the low group would have a higher credibility rating than a person picked at random from the control group ([26]). There was no statistically significant difference between the control group and the high-risk group (*p* = 0.420, *d* = 0.16).

Additionally, we assessed whether there was a correlation between the credibility of the defendant’s claim of innocence and the verdict. A statistically significant correlation was found (*r* = 0.514, *p* < 0.001). To further explore this, we conducted an exploratory *t*-test to examine potential differences in credibility between those who rendered a different verdict. A Welch’s *t*-test was conducted, given that the variances of the groups were unequal. The defendant’s claim of innocence was perceived as more credible by those who found the defendant not guilty (*M* = 5.05, *SD* = 1.48), compared to those who found the defendant guilty (*M* = 3.19, *SD* = 1.32; *t*(149.24) = −10.85, *p* < 0.001, *d* = 1.33). The absolute difference between the ratings on the credibility scale was on average almost two points for those who found the defendant guilty and those who found him not guilty. With a Cohen’s *d* of 1.33, 90.8% of those who found him not guilty will have credibility ratings above the mean of those who found him guilty. Moreover, there is an 82.7% chance that a person picked at random from the “not-guilty” group will be associated with a higher credibility rating than a person picked at random from the “guilty” group ([26]).

## 4. Experiments 2 and 3

After Experiment 1, two changes were introduced to improve the ecological validity of the experiment. First, the manner in which the risk assessment information was formulated in the vignette was individualized to better resemble the risk information in actual reports (see Section 2). Second, in addition to the Dutch student population in Experiment 2, Experiment 3 was conducted among a jury-eligible Flemish population (i.e., from the Flemish part of Belgium). This increases population validity of those experiments because these are individuals who can be invited to participate in a Flemish court of assizes.

Ethical approval for Experiments 2 and 3 was granted by the Social and Societal Ethics Committee of KU Leuven (reference: G-2023-6252) and the Committee of Ethics and Data of Leiden Law School (reference: 2023-01). The experiment was preregistered on the Open Science Framework (OSF; https://osf.io/sgeac) and all materials and data are available at OSF (in Dutch; https://osf.io/avz4h/).

### 4.1. Participants

A priori power analysis was conducted to calculate the required sample size for chi-square analysis. Using G*Power version 3.1.9.7 ([18]), with a power of 0.80, α = 0.05, *w* = 0.18 and two degrees of freedom, we calculated that a sample of 298 participants was needed. As in Experiment 1, we based the Cohen’s *w* effect size on the effect that was found by [50] ([50]) when they examined the impact of recidivism risk information in pre-sentence reports on verdicts (i.e., φ = 0.18). Similarly, we conducted a priori power analysis to calculate the required sample size for a multivariate analysis of variance. Although there have been no similar prior studies, [50] ([50]) have assessed the impact of a forensic mental health evaluation on the evaluation of evidence. They found a (non-significant) effect size, (*F*(3, 196) = 1.292, *V* = 0.019, *p* = 0.278). Using this effect size as reference, a sample size of 303 is needed (G*Power: MANOVA, global effects; *F*2(V) = 0.02, α = 0.05, power = 0.80, three groups, two response variables). Participation was voluntary and anonymous, and participants did not receive any incentive in exchange for participation.

### 4.2. Experiment 2: Dutch Sample

The participants in Experiment 2 were students from the law faculty at Leiden University. These students are future legal professionals. Participants were recruited during lectures and on the university’s virtual learning environment, as well as on social media platforms. The study fully took place online using Qualtrics and the link was shared during the in-class and online recruitment. Participation required a maximum of 15 min. In total, 264 participants completed the survey. Of those, 28 participants (10.6%) failed the manipulation check and/or three or more attention checks. The final sample consisted of 236 participants: 81 in the group with no risk information, 80 in the low-risk group, and 75 in the high-risk group. Participants were mainly female students (*n* = 206; 87.3%; male: *n* = 27; 11.4%; non-cis: *n* = 3; 1.3%) who studied criminology (*n* = 205; 86.9%; law: *n* = 12; 5.1%; both: *n* = 19; 8%). The majority of the sample (*n* = 229; 97.0%) were undergraduate students, with an average age of 20.6 years (*SD* = 3.8, *Mdn* = 20, *Min* = 17, *Max* = 68).[note 4]

### 4.3. Experiment 3: Flemish Sample

Participants in Experiment 3 were jury-eligible Belgians. The inclusion criteria were based on the eligibility criteria for jury candidacy: (1) included in the electoral register, (2) enjoying Belgian civil and political rights, (3) between 28 and 65 years old, and (4) can read and write in Dutch. In accordance with jury eligibility practices, politicians, judges, prosecutors, and high-ranking government officials were excluded. Participants were asked to indicate whether each criterion was applicable. When a criterion was not met, the survey was ended. People who have received (prison or community) sentences of a certain duration were also excluded from jury eligibility; however, for privacy reasons, this sensitive information was not obtained. Participants were recruited by flyers in public spaces and through Facebook groups and WhatsApp groups, while relying on a snowball effect by sharing the link to the Qualtrics survey.

In total, 85 participants who fulfilled the criteria for being jury-eligible completed the survey. Of those, 10 participants (11.8%) failed the manipulation check and/or three or more attention checks. The final sample consisted of 75 participants: 27 in the group with no risk information, 26 in the low-risk group, and 22 in the high-risk group. Participants were mainly female (*n* = 53, 70.7%; male: *n* = 21, 28.0%; non-cis: *n* = 1, 1.3%), with an average age of 39.5 years (*Mdn* = 37, *Min* = 28, *Max* = 58). The majority had finished a professional bachelor’s degree (*n* = 30; 40.0%) and worked full time (*n* = 60; 80.0%) in a service-providing occupation (*n* = 21; 28.0%).[note 5] Most participants had no prior experience with the criminal justice system (*n* = 54; 72.0%).

### 4.4. Results

Across both experiments, there were no statistically significant differences between the experimental conditions in terms of gender, age, and year of academic study (Experiment 2), and level of education, experience with the criminal justice system, and job situation (Experiment 3). Across all conditions, the observed conviction rate indicated that the evidence in the case was sufficiently ambiguous to cast at least some doubt on the defendant’s guilt (see Table 1; [9]; [50]). We expected that year of study (Experiment 2), education level, and experience with the criminal justice system (Experiment 3) might affect the various decisions independent of the manipulation. Bivariate analyses showed no significant correlation between year of study and the evaluation of evidence (witness: *r*(234) = −0.08, *p* = 0.210, line-up: *r*(234) *=* 0.07, *p* = 0.289), the convictionof guilt (*r*(234) = −0.07, *p* = 0.296), or the verdict (*χ*^2^ (4) = 2.69, *p* = 0.611, Cramer’s *V* = 0.611). Similar results were found for correlations between education level and the evaluation of evidence, and conviction of guilt and the verdict: *r*(73) = 0.001, *p* = 0.996 (witnesses); *r*(73) = −0.08, *p* = 0.513 (line-up); *r*(73) = 0.14, *p* = 0.247 (conviction of guilt; *χ*^2^ (4) = 2.68, *p* = 0.612, Cramer’s *V* = 0.189 (verdict). Finally, no significant correlations were found between experience with the criminal justice system and the dependent variables: *r*(73) = 0.09, *p* = 0.429 (witness evidence); *r*(73) = 0.16, *p* = 0.179 (line-up evidence); *r*(73) = 0.08, *p* = 0.482 (conviction of guilt); *χ*^2^ (1) = 0.25, *p* = 0.618, Cramer’s *V* = 0.058 (verdict). As such, these factors were not included as covariates in further analyses.

#### 4.4.1. Effect of Recidivism Risk on Evaluation of Evidence

To test the hypothesis that the witness statements and the line-up identification were considered more incriminating when the reported risk level was high compared to low, a multivariate ANOVA was preregistered for each sample.

We expected the evaluations of the two pieces of evidence to correlate (strongly) with each other across all conditions. Since this was not the case in Experiment 2 with Dutch students (Pearson’s *r* = 0.09, *p* = 0.194), we conducted separate ANOVAs, instead of the preregistered MANOVA test, for this experiment. For Experiment 3 with Flemish jury-eligible laypeople, both pieces of evidence correlated moderately to strongly (*r* = 0.441, *p* < 0.001) so a MANOVA test was conducted.[note 6] All three conditions were compared to assess whether the conditions with low or high risk differed from the control condition without risk information. For both experiments, the tests were not statistically significant. There was no evidence that the level of risk had any impact on the evaluation of the various pieces of evidence (Experiment 2 witness evidence: *F*(2, 233) = 0.561, *p* = 0.571, partial *η*^2^ = 0.005. Experiment 2 line-up evidence: *F*(2, 233) = 0.253, *p* = 0.776, partial *η*^2^ = 0.002; Experiment 3: *F*(4, 142) = 0.63, *p* = 0.643; Wilk’s *Λ* = 0.97, partial *η*^2^ = 0.02, see Table 2). Since the ANOVAs and MANOVA were not significant in both experiments, no further post hoc tests were conducted to assess differences between the low-risk and high-risk conditions. Participants in Experiment 3 (Flemishsample) reported significantly lower ratings for the evidentiary strength of the witness statements (*M* = 5.6, *SD* = 2.1) compared to participants in the Dutch sample (*M* = 6.2, *SD* = 1.9; *t*(309) = 2.24, *p* = 0.025, *d* = 0.30), but line-up identification ratings were similar.[note 7]

#### 4.4.2. Effect of Recidivism Risk on Conviction of Guilt

To test the hypothesis that participants are more convinced of the defendant’s guilt when a high level of risk is reported compared to a low level of risk, an ANOVA was performed in both experiments. First, we compared all three conditions to test whether the conditions with low or high risk differed from the condition without risk information. There were no statistically significant differences between the conditions: *F* (2, 233) = 0.259, *p* = 0.772, *η*^2^ = 0.002 for Experiment 2, and *F* (2, 72) = 0.22, *p* = 0.804, *η*^2^ = 0.006 for Experiment 3. Since the results of the ANOVAs were not significant, no further post hoc tests were conducted to assess differences between the low-risk and high-risk condition. Participants in Experiment 3 (Flemish sample) did report significantly lower scores for the conviction of guilt (*M* = 62.1, *SD* = 21.0) compared to participants in the Dutch sample (*M* = 68.6, *SD* = 16.7; *t*(105.6) = 2.43, *p* = 0.017, *d* = 0.36, see Table 3).[note 8]

#### 4.4.3. Effect of Recidivism Risk on the Verdict

To test the hypothesis that the level of recidivism risk affects the likelihood of a guilty verdict, two chi-square tests were conducted for each experiment. We compared all three conditions to test whether the conditions with low or high risk differed from the control condition without risk information (see Table 1). There were no statistically significant differences between any of the conditions: *χ*^2^ (2) = 0.06, *p* = 0.972, Cramer’s V = 0.016 for Experiment 2, and *χ*^2^ (2) = 0.949, *p* = 0.622, Cramer’s V = 0.112 for Experiment 3. As an exploratory (not preregistered) analysis, odds ratios were calculated to provide insight into the strength of the effects, even though they are not significant. In Experiment 2 (Dutch students), participants in the high-risk group were slightly less likely to receive a guilty verdict than those in the group with no information about risk (OR = 0.93, 95% CI [0.49, 1.76]) and those in the low-risk group (OR = 0.94, 95% CI [0.50, 1.80]). Participants in the low-risk group were slightly less likely to receive a guilty verdict, compared to those in the group with no information about risk (OR = 0.98, 95% CI [0.52, 1.84]). In Experiment 3 (Flemish laypeople), participants in the high-risk group were slightly more likely to receive a guilty verdict than those in the group with no information about risk (OR = 1.18, 95% CI [0.37, 3.73]) and less likely than those in the low-risk group (OR = 0.69, 95% CI [0.22, 2.18]). Participants in the low-risk group were slightly more likely to receive a guilty verdict, compared to those in the group with no information about risk (OR = 1.70, 95% CI [0.57, 5.09]). Participants in the Flemish sample reported a significantly lower proportion of guilty verdicts (42.7%) compared to participants in the Dutch sample (59.7%; *χ*^2^ (1) = 6.73, *p* = 0.010, Cramer’s V = 0.147).[note 9]

#### 4.4.4. Effect of Recidivism Risk and Evaluation of Evidence

We hypothesized that the reported level of recidivism risk would influence the evaluation of the evidence, which in turn would affect how convinced one is of the defendant’s guilt, as well as whether the decision-maker renders a guilty verdict. However, because there was no evidence that the level of risk had any impact on the evaluation of the various pieces of evidence, this mediation model was not further tested. We assessed the direct effects of recidivism risk and evaluation of evidence on both dependent variables, conviction of guilt and guilt judgment, via two separate regression models.

**Conviction of Guilt**. In Experiment 2, the linear regression model was statistically significant, *F*(4, 231) = 37.21, *p* < 0.001, *R*^2^ = 0.39. However, only the perceived incriminating nature of the witness testimonies (B = 4.09, *p* < 0.001) and of the line-up evidence (B = 3.76, *p* < 0.001) positively predicted conviction about guilt. There was no effect of the level of risk. In Experiment 3, the linear regression model was also statistically significant, *F*(3, 71) = 47.38, *p* < 0.001, *R*^2^ = 0.67. Again, only the perceived incriminating nature of the witness testimonies (B = 3.15, *p* < 0.001) and of the line-up evidence (B = 6.13, *p* < 0.001) predicted the level of conviction about guilt. There was no effect of risk information.

**Guilty Verdict**. In Experiment 2, the logistic regression model was statistically significant, *χ*^2^(4) = 57.35, *p* < 0.001. The model explained approximately 29.1% (Nagelkerke *R^2^*) of the variance in assigned verdicts and correctly classified 77.1% of cases. Similar to the linear regression model, the more incriminating someone judged the evidence to be (both for the witness testimonies, B = 0.47 *p* < 0.001, and the line-up evidence, B = 0.41, *p* < 0.001), the higher the likelihood was that the person would find the defendant guilty. There was no significant effect of risk information. In Experiment 3, the logistic regression model was also statistically significant, *χ*^2^(4) = 50.30, *p* < 0.001. The model explained approximately 65.6% (Nagelkerke *R^2^*) of the variance in assigned verdicts and correctly classified 80.0% of cases. Similar to the linear regression model, the more incriminating someone judged the evidence to be (both for the witness testimonies, B = 0.54, *p* = 0.011, and the line-up evidence, B = 1.11, *p* < 0.001), the higher the likelihood was that the person would find the defendant guilty. There was no significant effect of risk information.

## 5. Discussion

The present experiments were the first to assess the potential context effect of the reported level of recidivism risk on the evaluation of evidence and defendant credibility, as well as the determination of guilt. Before interpreting the findings of each experiment, one interesting observation relates to the differences in conviction rates between the three experiments despite the vignettes being very similar. In Experiment 1, with Flemish students (*N* = 368), 74% of participants found the defendant guilty, while this was 60% in Experiment 2 with Dutch students (*N* = 256), and 43% in Experiment 3 with a jury-eligible Flemish sample (*N* = 75). The last group seemed especially reluctant to convict the defendant in this assault case with ambiguous evidence. Prior studies that used this vignette found a conviction rate of 77% among 51 Dutch judges ([9]) and 82% among 200 Dutch students ([50]). Despite these varying rates of conviction, our findings with respect to the effect of risk information are consistent across the experiments.

In each experiment, we hypothesized that decision-makers would be more likely to convict a defendant and be more convinced of his guilt when they read in the case file that the defendant has a high risk of recidivism compared to a low risk. Our argument for this hypothesis was that reading about a high recidivism risk would activate a ‘guilt schema’ through which the court file would be interpreted ([7]; [53]). As such we also hypothesized that the ambiguous evidence would be evaluated as more incriminating in the high-risk condition than the low-risk condition. Similarly, the defendant’s claim of innocence was hypothesized to be perceived as less credible in the high-risk condition compared to the low-risk condition. However, a high recidivism risk did not create a biasing context effect relating to decision-making regarding guilt, either directly or via the evaluation of the presented evidence. Even though risk information did affect the credibility of the defendant’s innocence claim (Experiment 1), these findings suggest that, in the context of our experimental design, the inclusion of recidivism risk information does not influence participant’s judgments of guilt. One possible interpretation could be that participants were able to appropriately distinguish between information about an individual’s past actions and predictions of their future behavior. However, this would contrast with established cognitive biases, such as the fundamental attribution error, which suggests that people tend to overemphasize dispositional factors while underestimating situational influences when interpreting others’ behaviors ([38]). When combined with a general consistency bias—the tendency to expect behavioral stability over time—such biases typically increase reliance on past behavior as a predictor of future actions ([40]). However, to our knowledge, there is a lack of empirical research addressing whether the reverse also holds: that is, whether information about an individual’s potential future behavior influences retrospective judgments about their past conduct.

On the other hand, there may be methodological explanations for this null finding on evidence evaluation and judgments of guilt. First of all, some of the acquired samples were smaller than the calculated required sample size of 300 participants. Experiment 2 included 256 participants and Experiment 3 had 75; therefore, the last experiment in particular lacked the power necessary to detect the effect if there had been one. However, we repeated all the analyses on the combined samples of Experiments 2 and 3 (*N* = 311) to assess this power problem. These combined analyses yielded results similar to those of the individual analyses, which supports the null findings in this study. Furthermore, the effects of evidence evaluation on decisions about guilt in the regression models showed that the perceived incriminating nature of the evidence contributed significantly to the verdict, which supports the assumption that decision-making about guilt was done properly. Participants did not seem to use predictions of future behavior to evaluate past actions. In what follows, we reflect on other explanations that may have contributed to the null findings: the case vignette, the timing of the risk information in the vignette, and how the risk information was reported.

### 5.1. Case Vignette

One possible explanation for our null findings is related to the case vignette, and particularly to the relatively low conviction rates and the related (neutral) evidence evaluation across the different conditions, irrespective of the addition of risk information (especially in Experiments 2 and 3). These findings indicate that participants were not easily convinced of the defendant’s guilt. The addition of risk information did not change this belief. The case vignette originated from a study about the ‘conviction paradox’: in case of a serious offense, the standard of proof applied by a judge may be lower than for less serious, but comparable offenses ([9]). Even though the case vignette in the present experiments was a serious offense, it is possible that risk information may only yield contextual bias for the most severe crimes, such as rape and homicide. Future research could explore this issue.

### 5.2. Timing of the Risk Information

Another possible explanation lies in the timing of the risk information within the vignette. In our study, the risk information was presented at the very end, after the evidence against the suspect was provided. The existing literature has shown that the final piece of information in a sequence can be disproportionately influential, as seen in recency effects ([25]; [35]; [58]). Our hypothesis was that the risk information would serve as a framework through which the evidence would be evaluated, rather than being treated as evidence itself. The null finding may suggest that this framework did not have the anticipated impact because it was presented last in the sequence. It is possible that participants’ processing of the evidence was already concluded by the time the risk information was introduced, thus preventing any retrospective influence on their evaluations.

This interpretation aligns with the work of [3] ([3]), which demonstrated how order effects can influence the evaluation of evidence. Their research suggests that when strong incriminating evidence is presented first, ambiguous subsequent information is processed more superficially and quickly. In contrast, when ambiguous evidence is presented first, without a defined belief of guilt or innocence, the evidence is processed more deeply, and subsequent information is less likely to alter the initial evaluation. In our study, the sequence of presenting the evidence before the risk assessment information may have led participants to evaluate the evidence in a relatively deep, unbiased manner, rendering the risk information ineffective in altering that initial evaluation. This is supported by the relatively neutral evaluation of the evidence in the case (around 6 on a scale of 0–10). Yet some studies have found that assimilation of evidence can also occur backwards, i.e., when decision-makers are asked to evaluate the evidence after they have received all the information. [36] ([36]) found a context effect of psychiatric history of the defendant on evidence evaluation and conviction about guilt. Information about the defendant’s psychiatric history was presented after participants had been informed about the evidence. Yet they gave their evaluation of evidence after they had learned about the psychiatric history, thereby showing a bilateral assimilation effect: processing seemingly incriminating information (i.e., the defendant’s psychiatric history) makes evidence already processed gain in perceived strength when there is already a coherent representation of the case based on contextual information ([41]).

One limitation of the current study, therefore, is that we did not experimentally manipulate the order in which the risk information was presented. Given the potential for order effects, it would be valuable for future research to examine whether presenting the risk information first might lead to different results than presenting it last. For instance, when the recidivism risk is provided upfront, participants may more strongly rely on this as frame of reference for evaluating the evidence that follows. This could potentially create a stronger context effect, as suggested by previous research on order effects in decision-making ([3]; [1]). To further test this effect, a mediation analysis would be appropriate.

### 5.3. Presentation of Recidivism Risk Information

Additionally, the risk information, in this experimental design, may not be a sufficient contextual cue to affect decision-making. This could be due to a variety of factors, including how the risk information was reported, participants’ expectations about risk assessment (e.g., reliability, validity), or how well they understood the risk information. First, although we adjusted how future risk was reported after the first experiment, making it more closely resemble an actual risk report, the paragraph remained shorter and less detailed than risk reports typically found in real case files ([30]). The current information may not have been sufficiently convincing to function as a frame through which the other information is perceived and processed. Moreover, we have not assessed the participants’ understanding of risk of recidivism and the terminology used to describe this. The student sample was expected to have some understanding of the concept through their education; however, we did not assess this explicitly. Future research could evaluate the participants’ understanding of recidivism risk. For instance, several statements on recidivism and culpability could be added in the last phase of the experiment, after the participants have answered questions about the evidence and guilt.

### 5.4. Effect on Credibility

That said, the risk information was sufficient to impact the credibility of a defendant’s claim of innocence. Participants who read that the defendant had a low risk of recidivism found his claim of innocence more credible than participants who read that the defendant was high risk or who received no risk information at all. This finding differs from our expectation that the high-risk information would have a biasing effect. Rather than high risk having an aggravating effect, information about low recidivism risk seems to have a mitigating effect on perceived credibility. This is, in fact, in line with the sentencing study of [10] ([10]), in which low-risk judgments were linked to shorter prison sentences, compared to high risk and no risk information. Information about low risk may create a frame through which the defendant looks less guilty. Overall, the finding that recidivism risk information affects credibility ratings is in line with prior findings that contextual irrelevant information, such as suspect emotionality or nonverbal behavior, can affect the credibility assessment of a suspect’s statement (e.g., [2]; [39]).

Furthermore, credibility ratings correlated meaningfully with judgments of guilt. That is, the more credible the participants rated the defendant’s claim of innocence to be, the more they were inclined to reach a not-guilty verdict in Experiment 1. An average difference of two points on a 10-point credibility scale may be considered noteworthy. Moreover, the average rating of the group that rated the defendant as not guilty was above 5 (*M* = 5.05). This means that, on average, this group judged the claim of innocence as credible, whereas the other group, who found the person guilty, typically found the defendant’s claim not credible (*M* = 3.19). In sum, the difference between the two groups is not only significant but also meaningful. This is in line with research on perceived trustworthiness of defendants and its impact on the determination of guilt. [34] ([34]), for example, showed that for an untrustworthy defendant, fewer pieces of (ambiguous) evidence were required to arrive at a guilty verdict than for a trustworthy defendant. An analogue process may have occurred in our experiment: defendants who were said to be low risk may have been judged to be more credible and this, in turn, may have increased the likelihood of being found not guilty.

### 5.5. External Validity

Experimental studies on criminal justice decision-making excel in internal validity but face challenges with population validity and ecological validity ([6]). While our Belgian jury-eligible sample represents actual jurors, our student samples differ from professional judges in experience and training. Recent studies confirm that law students are not very good proxies for professional judges (see [45]; [47]). Despite these limitations, our approach remains theoretically relevant. We assert that the influence of recidivism risk information represents a fundamental cognitive phenomenon that may transcend professional expertise ([20]; [49]). The use of students and jury-eligible participants, therefore, allows us to examine how biases and context effects may manifest, following the assumption that mechanisms underlying these effects are fundamental and universal to human decision-making. In fact, our direct comparison across decision-maker types, rarely undertaken in previous research, revealed similar decision-making across populations. Nevertheless, to address validity concerns, future research could explore more immersive or ecologically valid methodologies, such as virtual reality simulations of courtroom environments (for methodological standards, see [56]), and experiments with real judges or jurors. That said, even within the constraints of this study, we believe our study provides valuable first steps into studying potential biases introduced by risk assessment information in legal contexts. As such, while the ecological validity of our study may be limited, we believe it lays a foundation for future research that can build on these findings and explore their relevance to real-world legal decision-making. If an effect of recidivism risk is found in future studies, it could have significant practical implications for legal systems that lack procedural guidelines on when such information should be considered. Demonstrating a context effect could provide empirical support for reforms in countries like Belgium and the Netherlands. Conversely, if no effect is found in future studies with legal decision-makers, this would highlight the resilience of judicial decision-making processes, suggesting that triers of fact are capable of disregarding recidivism risk information when it is legally irrelevant.

## 6. Conclusions

In some countries, such as Belgium and the Netherlands, risk assessment reports are provided to triers of fact at the outset of a trial. Similarly, in the US, reports on recidivism risk may be available during plea negotiations between prosecutors and defense attorneys. While recidivism risk information is formally irrelevant during trial proceedings until post-conviction, its presence may nevertheless contaminate pre-sentencing decision-making. This study represents a first attempt to test such context effects of recidivism risk information on such decisions. We did not find support for the hypothesized effects on evidence evaluation and judgments of guilt. Nevertheless, recidivism risk information did impact the perception of the defendant’s credibility. More specifically, learning that a defendant was determined to have low risk of recidivism increased the credibility of his claim of innocence compared to when the defendant was high risk or when risk information was not provided. Moreover, the credibility ratings correlated with the decision about guilt: when participants believed more in a defendant’s claim of innocence, they were more likely to find the defendant not guilty. As such, this study provided some preliminary indications that the presence of risk assessment information in a criminal file could result in a context effect on legal decisions. Despite the mixed findings, the methodological considerations raised by our study offer valuable directions for future research. This is particularly relevant given the prominent use of forensic risk assessments in criminal trials in Western countries ([43]; [52]). Specifically, there is a need for further investigation into the timing and presentation of risk information. To strengthen ecological validity, future replications of this research should incorporate more immersive or real-world designs.

## Figures and Tables

**Table 1 behavsci-15-01277-t001:** Proportions of guilty verdicts.

	Experiment 2 (*N* = 236)			Experiment 3 (*N* = 75)		
	Total	Guilty	%	Total	Guilty	%
No information	81	49	60.5	27	10	37.0
Low risk	80	48	60.0	26	13	50.0
High risk	75	44	58.7	22	9	40.9
Total	236	141	59.7	75	32	42.7

**Table 2 behavsci-15-01277-t002:** Evaluation of the evidence in Experiments 2 and 3.

	Experiment 2 (*N* = 236)		Experiment 3 (*N* = 75)	
	Witness Statements *M* (*SD*)	Line-Up Identification *M* (*SD*)	Witness Statements *M* (*SD*)	Line-Up Identification *M* (*SD*)
No information	6.4 (1.7)	6.4 (1.9)	5.6 (2.3)	6.2 (1.9)
Low risk	6.1 (2.0)	6.2 (1.6)	5.9 (2.0)	5.9 (2.3)
High risk	6.1 (1.9)	6.2 (1.9)	5.5 (1.9)	6.6 (2.2)
Total	6.2 (1.9)	6.3 (1.8)	5.6 (2.1)	6.2 (2.1)

**Table 3 behavsci-15-01277-t003:** Conviction of guilt in Experiments 2 and 3.

	Experiment 2 (*N* = 236) *M* (*SD*)	Experiment 3 (*N* = 75) *M* (*SD*)
No information	68.1 (18.0)	60.3 (19.2)
Low risk	69.7 (16.8)	62.3 (22.7)
High risk	67.9 (15.4)	64.3 (21.6)
Total	68.6 (16.7)	62.1 (21.0)

## Data Availability

This study was preregistered on the Open Science Framework (OSF; Experiment 1: https://osf.io/ce4u7; Experiments 2 and 3: https://osf.io/sgeac), and materials, data, and syntax can be accessed at the study’s OSF project page (https://osf.io/avz4h/).

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
