# Peer review of "Justice at Risk? The Influence of Recidivism Risk Information on Evaluation of Evidence and Determination of Guilt"

_behavsci, 2025, doi:10.3390/bs15091277_

Round 1

Reviewer 1 Report

Comments and Suggestions for Authors

I enjoyed reading this paper and found it extremely interesting, insightful and topical. I particularly appreciated the empirical studies and the use of the ANOVA method, and encourage the authors to expand on their findings and continue their empirical work. I did not detect any serious methodological, structural or substantial flaws that would require additional intervention from the authors. I think this article can be published as is.

As a follow-up to this study: it would be interesting if the authors included recidivism-predicting technologies in their future research: as is well known in AI circles, automated recidivism predictions have given way to much academic debate and some court practice (e.g. the Loomis case). I wonder what would happen if the case vignettes (mentioned in the reviewed article) included automated predictions and asked respondents to - inter alia - flag their levels of trust in the accuracy of those predictions... But this may not at all be the direction the authors wish to take, which is, of course, perfectly fine.

The bottom line is: this article is publishable in its current state.

Author Response

Summary

I enjoyed reading this paper and found it extremely interesting, insightful and topical. I particularly appreciated the empirical studies and the use of the ANOVA method, and encourage the authors to expand on their findings and continue their empirical work. I did not detect any serious methodological, structural or substantial flaws that would require additional intervention from the authors. I think this article can be published as is.

As a follow-up to this study: it would be interesting if the authors included recidivism-predicting technologies in their future research: as is well known in AI circles, automated recidivism predictions have given way to much academic debate and some court practice (e.g. the Loomis case). I wonder what would happen if the case vignettes (mentioned in the reviewed article) included automated predictions and asked respondents to - inter alia - flag their levels of trust in the accuracy of those predictions... But this may not at all be the direction the authors wish to take, which is, of course, perfectly fine.

The bottom line is: this article is publishable in its current state.

Response: We thank the reviewer for this thoughtful and positive review. We are glad that the reviewer enjoyed reading our manuscript and values our empirical approach. We appreciate the reviewer’s ideas and suggestions for follow-up studies; they are interesting avenues to extend this work! We are grateful for this input and for the endorsement of our work.

Reviewer 2 Report

Comments and Suggestions for Authors

The paper presents an experimental examination of the influence of a defendant’s risk assessment on the decision on guilt, and other legally relevant elements of courtroom decision-making, such as evaluation of evidence and defendant’s credibility. Quite surprisingly, the main result seems to support the null hypothesis. Some other results are very interesting as well, and the correlation between higher credibility rating and non-guilty verdict can be treated as evidence that experimental manipulation worked in general.

The null result should not be treated as less valuable here – on the contrary, my impression is that in the field of studying the relation between evidence, decisions on guilt and sentencing there exists a solid drawer effect. The hypotheses are grounded in the theory very well. The lack of influence of irrelevant factors in the form of risk assessments despite the strong biasing potential is of course good news for the legal system and – as the Authors rightly point out – should be studied further, especially on judges in continental systems (although, as Authors argue with insight in Section 1.5, the paper should be interesting for common law readers as well). I wholeheartedly appreciate the links to preregistration and study materials (although I was not able to access the latter, please make it public).

I believe that the article in its current form is close to being publishable, nonetheless I enclose some (more or less nitpicky) comments below.

  • l. 32-33 – is the first sub-section named exactly the same as the whole article? That is inadequate, the title of the subsection could be something like “risk assessment”.
  • l. 38-39 – I would say that not only the risk assessment can influence the decision on pretrial detention, but is an essential part of this type of decision – unless the Authors mean here a given set of tools used by non-judicial actors (which they describe in l. 43-47) rather than a concept of risk assessment in general. If so, this should be made more clear and precise.
  • l. 56 – for clarity’s sake, I would add “other legal decision in criminal courts, in which it is not a relevant factor” or something like that, to underline the difference between the contexts described earlier in which the risk assessment is a relevant element of a given decision (see also my comment to the l. 79).
  • l. 59-77 – the whole section is a little bit messy when it comes to systematization of biases. Aren’t all cognitive biases contextual? The effects described in l. 70-73 do not sound like cases of confirmation bias (as the expert changed their mind), but rather Asch’s style conformity or bandwagon effect..
  • l. 79 – while risk assessment is relevant to sentencing in most cases in modern Western legal systems, philosophically, an old school retributivist could argue that the just punishment should be backwards-looking (focusing on the deed rather on the person, ie. the Tatstrafrecht instead of Täterstrafrecht) and thus any forecast should be irrelevant at the sentencing stage.
  • l. 105-106 – it would be worth adding that the study had experimental character.
  • l.121-123 – wouldn’t that be – more simply and in general –  a case of halo effect (in the negative, “horns” version)? “The risky defendant is a bad person, bad people commit crimes”. Or, as the Authors describe later (l. 652-655), a fundamental attribution error?
  • generally, literature review could be more thorough (while I acknowledge that the article is an original experimental piece). For example, the review misses a seminal article on disregarding inadmissible evidence by judges (Wistrich, Guthrie, Rachlinski 2005, Can Judges Ignore Inadmissible Information - The Difficulty of Deliberately Disregarding, 153 U. Pa. L. Rev. 1251; and for a more theoretical context see Lagnado, Harvey 2009, The Impact of Discredited Evidence, Psychonomic Bulletin & Review)
  • “bias snowball” etc.  – l. 149- 151; l. 210-212 – it may also be the other way around. The initial assessment of guilt may influence the belief in the credibility of the evidence. The Authors return to this issue in l. 706-718. The mediation analysis could be of some help here, but since the null result was obtained (which Authors explain on l. 596-600), this remains a subject for future research.
  • l. 569-571; 591-593 – it would be worth reminding the readers that the Flemish sample was composed of laypeople and Dutch samples were composed of the law/criminology students. As the results for the Flemish sample were in line with the hypotheses, this is also a point worth touching in more detail in the Discussion section. Could it be that legal education somehow debiases people? Or is it the life experience of laypeople that influenced the biasing element of risk assessment? Or the difference lies in differences in (legal) culture between Belgium and the Netherlands (but that is less probable as both student groups were similar)?
  • l. 658-660 – this is an extremely interesting observation for future studies. Mechanisms similar to hindsight and self-fulfilling prophecies may work in that context, but that is to be tested in another study.
  • l. 688-705 – serial position effects are tricky beasts, as one can argue over the strength of recency/primacy for hours. It seems however, that recency prevails in this context (to reference a few more studies not included, but very relevant: Enescu, Kuhn 2012, Serial effects of evidence on legal decision-making, The European Journal of Psychology Applied to Legal Context; Maegherman et al.  2021, Law and order effects: on cognitive dissonance and belief perseverance, Psychiatry, Psychology and Law; Wojciechowski et al. 2025, Order Effects and the Evaluation Bias in Legal Decision Making, Decision), although, I agree with the Authors that there may be a case for primacy effect here (combined with framing/confirmation bias, cf. also Pennington 2006, Witnesses and Their Testimony: Effects of Ordering on Juror Verdicts, Journal of Applied Social Psychology).
  • l. 754-815 – the whole Section is overly lengthy in my opinion. The limitations of experimental studies in legal decision-making, including student sample issues, are pretty well known. A reference and commentary with regard to Spamann and Klöhn’s study (2024; Can Law Students Replace Judges in Experiments of Judicial Decision-Making? Journal of Law & Empirical Analysis, 1(1), 149-161) would be worthwhile instead. Another interesting study (comparing law students and laypeople) that is of relevance is Tobia’s article on legal expertise (2024; Legal concepts and legal expertise, Synthese 203).

Reviewer 3 Report

Comments and Suggestions for Authors

Overall, a nicely done paper with a good flow and openness on how to contextualise the null findings yet still provide some useful advice. 

The small sample sizes were foreseeable to be a problem in terms of power, but cannot go back and change that.

A few bits that could be improved:

Line 70 refers to a case where forensic handwriting experts change their initial opinion after being told another examiner had a different conclusion. This is given as an example of contextual confirmation bias in a forensic setting. Yet more info would need to be given to make this connection because of its retrospective nature. If the reason the experts changed their initial opinion was because of new information gleaned from the other expert, then this would not necessarily represent bias versus willingness to be open to new information and thus to legitimately change one's professional judgement. 

The sentence beginning at line 89 indicates that Federal Rule of Evidence 404 prohibits character evidence and that risk assessment information counts as character evidence. I do not read the source given (Sampson & Smith) actually saying that, and do not believe it is true in any event. Risk information is offered to predict future behavior, not as evidence of past behavior. The reason (at least in the US) that risk assessment info would not generally be admissible for adjudication is that it is not relevant to an issue of whether the person committed the act alleged.

On that note, it would be helpful to a reader to understand why risk information is offered to a jury in Belgium (starting line 96)?

In the discussion in Section 5.2, consider whether another reason to explain the null findings is that the mock jurors are asked immediately to respond to the queries versus a real jury would have much more time to process the risk assessment information and thus to impact their judgements about guilt (around line 710). 

The paper does leave open some critiques about whether Experiments 2 and 3 could isolate to the future risk aspect versus the information in the paragraph (beginning line 283) about the person's past and current behavior and character. For example, it is not possible to know whether the mock jurors would react to info that the defendant smokes marijuana and hangs out in the street with troublemaking friends (suggesting a good fit to beat someone up) versus the assessment of high future risk of violence. Another issue is that some studies show different perceptions of risk information when just the judgement is given (e.g., high risk) as opposed to the judgement plus information on the individual factors that were assessed that drove that judgement. Thus, the risk condition in Experiment 1 is very different than in Experiments 2 and 3. 
